# ANQ: Approximate Nearest-Neighbor Q Learning

## Abstract

In specific domains such as autonomous driving, quantitative trading, and health-care, explainability is crucial for developing ethical, responsible, and trustworthy reinforcement learning (RL) models. Although many deep RL algorithms have attained remarkable performance, the resulting policies are often neural networks that lack explainability, rendering them unsuitable for real-world deployment. To tackle this challenge, we introduce a novel semi-parametric reinforcement learning framework, dubbed ANQ (**A**pproximate **N**earest Neighbor **Q**-Learning), which capitalizes on neural networks as encoders for high performance and memory-based structures for explainability. Furthermore, we propose the Sim-Encoder contrastive learning as a component of ANQ for state representation. Our evaluations on Mu-JoCo continuous control tasks validate the efficacy of ANQ in solving continuous tasks while offering an explainable decision-making process.

## 1 Introduction

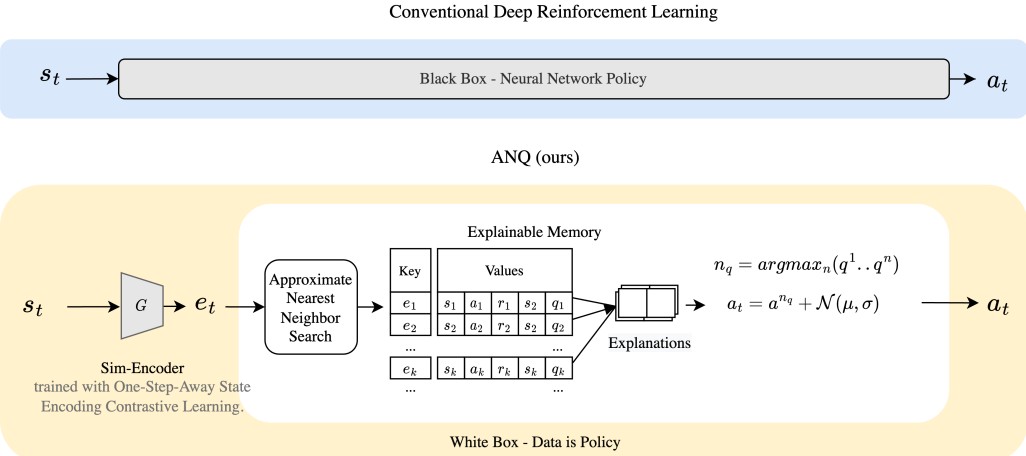

Figure 1: Overall Architecture of Our Approach

In recent years, parametric reinforcement learning methods featuring end-to-end training, such as Proximal Policy Optimization (PPO) [Schulman et al., 2017], Soft Actor-Critic (SAC) [Haarnoja et al., 2018], and Deep Deterministic Policy Gradient (DDPG) [Lillicrap et al., 2015], have gar-nered significant attention within the reinforcement learning community. These approaches have demonstrated remarkable success in addressing decision-making challenges across diverse domains,

including robotics [Hwangbo et al., 2019], video games [Mnih et al., 2015], and board games [Schrittwieser et al., 2020]. Nevertheless, the incorporation of deep neural networks in these methods presents a major obstacle to interpreting the underlying rationale of their decision-making processes. This limitation hampers the application of such methods to numerous real-world scenarios, such as autonomous driving [Kiran et al., 2021], quantitative trading [Zhang et al., 2020], and beyond. Consequently, further investigation is necessary to enhance the interpretability and practical utility of these reinforcement learning techniques in complex, real-world contexts.

This issue calls for the research of explainable reinforcement learning (XRL) which aims at obtaining RL models that are both explainable and of high performance. Fidelity is one of the major objectives in XRL [Milani et al., 2022] which measures to what extent the model makes decisions following its explanation. Among different XRL algorithms, *white-box* algorithms (i.e., making decisions directly using explainable models such as linear models or decision trees) enjoys high fidelity than the others. (We defer the introduction of other XRL algorithms to Section 5.3.)

Memory-based reinforcement learning, following the non-parametric paradigm, is a popular class of white-box algorithm and differs from widely researched parametric methods in deep reinforcement learning. The approximation function in memory-based reinforcement learning is determined directly by the training samples, rather than relying on a gradually-updated parameterized function. Prominent memory-based methods include EC [Blundell et al., 2016], NEC [Pritzel et al., 2017], and EMDQN [Lin et al., 2018] (see more in Ramani [2019]). Memory-based reinforcement learning has several benefits, including being able to approximate a universal class of functions, the ability to directly impact the policy with newly accumulated data without back-propagation updates Blundell et al. [2016], the mitigation of the curse of dimensionality in global estimation Sutton and Barto [1998], and higher data sampling efficiency and faster learning Lin et al. [2018]. Most importantly, memory-based reinforcement learning possesses the advantage of improved explainability due to its human-understandable decision making system (i.e., the memory consists of pre-collected samples).

Despite its potential for self-explainability through white-box decision-making, the utilization of memory-based reinforcement learning for enhancing explainability remains relatively unexplored. Existing studies investigating the use of episodic memory for explanations, such as Cruz et al. [2019], Pritzel et al. [2017], Blundell et al. [2016], have been limited to grid world environments or discrete tasks. In contrast, our work aims to expand this research scope to encompass continuous robotics tasks in Mujoco by proposing a comprehensive memory-based self-explainable framework.

Efficiently retrieving relevant data from extensive databases presents a significant challenge in developing an effective memory-based reinforcement learning algorithm, particularly in continuous control tasks as emphasized by Sutton and Barto [1998]. However, recent advancements in approximate nearest-neighbor searching algorithms, such as Hierarchical Navigable Small World (HNSW) Malkov and Yashunin [2018], have demonstrated their effectiveness in swiftly retrieving pertinent information from billions of records in natural language processing (NLP) tasks. Such methods have been successfully applied to question-answering Kassner and Schütze [2020] and text generation Borgeaud et al. [2022] tasks. In addition to NLP applications, retrieval-based systems have been integrated with deep reinforcement learning algorithms, resulting in enhanced sample efficiency Goyal et al. [2022], Humphreys et al. [2022].

The contributions of our paper are summarized as follows:

- We introduce a novel framework, ANQ, which offers efficient control in continuous domains across a wide range of Mujoco experiments, while maintaining high explainability through its "data is policy" design principle.

- We present the Sim-Encoder, a nearest neighbor contrastive learning approach for state representation, which demonstrates its effectiveness in memory retrieval learning tasks.

## 2 Preniminaries

We first introduce notations and summarize the conventional episodic control method.

## 2.1 Notation

In this work, we study policy learning in continuous action space $\mathcal{A}$ and observation space $\mathcal{S}$. We consider a Markov decision process with transition $s_{t+1} \sim p(s_{t+1}|s_t, a_t)$. After performing an action, the agent receives a reward, and the ultimate goal is to optimize the policy to maximize returns.

A key-value-based dataset $\mathbb{D}$ stores the key as the state embedding $e$. The database consists of rows of $\{k, e_t, s_t, a_t, r_t, s_{t+1}, q_t\} \in \mathbb{D}$ and columns of $\{\mathbb{K}, \mathbb{E}, \mathbb{S}_t, \mathbb{A}, \mathbb{R}, \mathbb{S}_{t+1}, \mathbb{Q}\} \in \mathbb{D}$. $\mathbb{K}$ represents the set of all record IDs. The maximum number of rows is $M$. The observation Sim-Encoder network is denoted as $\mathbf{G}_\theta$ parameterized by the network parameters $\theta$.

For database operating, in total, six operations are defined in the memory module: APPEND, TRIM, GET, UPDATE, SEARCH, and INDEX. More corresponding explanations for these operations can be found in Sec.3.2.

## 2.2 Episodic Control

Episodic control methods enhance sampling efficiency and episodic returns by using an external memory database for interactions such as writing, reading, and updating. The concept was first introduced in Blundell et al. [2016], which resolved complex sequential decision tasks.

This method is defined for discrete spaces. It proposes the following Q table update mechanism:

$$Q^{EC}(s, a) = \max(Q^{EC}(s, a), R) \tag{1}$$

After the update, it generates an effective Q Table. During the policy execution phase, if an observation-action pair exists in memory, the Q value is retrieved directly from the table. However, if the pair is not found in memory, an approximation matching and estimation process is required. The agent queries the Q Table using the following approach to obtain the Q value.

$$\hat{Q}^{EC}(s, a) = \frac{1}{N} \sum_{n=1}^{n=N} Q(s^n, a) \tag{2}$$

The objective of episodic control is to accelerate learning speed and improve decision quality. An external memory module can then compensate for drawbacks such as low sample efficiency and slow gradient updates.

In previous discussions, the Episodic Control (EC) method has been investigated under both discrete actions and continuous actions (Li et al. [2023], Kuznetsov and Filchenkov [2021]). However, the explainability of EC in continuous action spaces suffers from low fidelity due to the utilization of a policy network. In this paper, we set out to achieve two objectives concurrently. First, we explore how Episodic Control can be effectively applied in continuous action spaces. Second, we strive to leverage the memory of Episodic Control to attain explainability benefits.

## 3 Method

The complete algorithm is presented in Algorithm 1, and the illustration of the inference pipeline can be observed (cf. Fig.1). The proposed method involves generating an embedding vector $e_t$ from the observation using the Sim-Encoder. Subsequently, we employ the HNSW algorithm Malkov and Yashunin [2018] to search for the nearest neighbor set $e^n$ within the memory. Each neighbor is associated with an action and a Q value, and the action with the highest Q value is selected as the policy output. It is worth noting that this action is continuous, which distinguishes it from previous EC work Blundell et al. [2016].

First, the Sim-Encoder in embedding observations into a cosine space is augmented via One-Step-Away State Encoding Contrastive Learning. This approach employs adjacent states as positive samples for contrastive learning, with experimental outcomes demonstrating that the implementation of the Sim-Encoder considerably enhances performance.

**Algorithm 1** ANQ Algorithm
___
**Input:**
Database $\mathbb{D}$
with each row notated as $\{k, e_t, s_t, a_t, r_t, s_{t+1}, q_t\} \in \mathbb{D}$
with each column notated as $\{\mathbb{K}, \mathbb{E}, \mathbb{S}_t, \mathbb{A}, \mathbb{R}, \mathbb{S}_{t+1}, \mathbb{Q}\} \in \mathbb{D}$
Observation Sim-Encoder network $\mathbf{G}_\theta$
Contrastive learning function CL
Gaussian distribution for action noise $\mathcal{N}(\mu, \sigma)$
**for** each iteration **do**
    **for** each environment step **do**
        $e_t = G_\theta(s_t)$
        $k_1..k_n =$SEARCH$(e_t)$
        $(a^1, q^1)..(a^n, q^n) =$GET$(k_1..k_n)$
        $n_q = argmax_n(q^1..q^n)$
        $a_t = a^{n_q} + \mathcal{N}(\mu, \sigma)$
        $s_{t+1} \sim p(s_{t+1}|s_t, a_t)$
        APPEND $( e_t, s_t, a_t, r_t, s_{t+1})$
    **end for**
    **for** sampled minibatch $\{s_t, s_{t+1}\}$ **do**
        $\mathcal{L}_\theta = CL(s_t, s_{t+1})$
        update networks $G_\theta$ to minimize $\mathcal{L}$
    **end for**
    $\mathbb{K}^1..\mathbb{K}^n =$SEARCH$(\mathbb{E})$
    **for** each learning step **do**
        $\mathbb{Q}^1..\mathbb{Q}^n =$ GET$(\mathbb{K}^1..\mathbb{K}^n)$
        $\hat{\mathbb{Q}} = \mathbb{R}_t + \gamma \frac{1}{N} \sum_{n=1}^{n=N} \mathbb{Q}^n$
        UPDATE$( \mathbb{Q}, \hat{\mathbb{Q}})$
    **end for**
    TRIM()
    INDEX()
**end for**
___

Subsequently, in order to acquire a comprehensive Q-table, we employ in-memory learning, which involves the batch computation of all Q-value estimations and Q-learning updates for each state stored in memory. The training process undergoes iterative cycles until the global Q-value converges.

### 3.1 Embedding Module

We introduce our novel approach, the "One-Step-Away State Encoding Contrastive Learning." The reason for using a one-step-away state as a positive sample is that the most informative actions and q-values for the current state are derived from a scenario that is most similar to it (Blundell et al. [2016]).

$$e_t = G_\theta(s_t) \tag{3}$$

This method aims to effectively represent the state with contrastive learning. Specifically, we utilize positive samples that consist of a state pair $s_t, s_{t+1}$ that are one step away. The resulting state representation is designed such that the nearest neighbor of each state is reachable within one step. We adopt a similar objective to SimCLR Chen et al. [2020], aiming to maximize the similarity between two vectors as measured by cosine similarity $sim(u, v) = u^T v/(|u||v|)$. The Sim-Encoder is a standalone component trained to maximize the similarity of embedding, without reward information but only state transition tuples.

$$\theta = argmax_\theta \mathbb{E}_{(s_t, s_{t+1}) \sim \mathbb{D}}[sim(G_\theta(s_t), G_\theta(s_{t+1}))] \tag{4}$$

Table 1: Memory Operations

| Operation | Description |
|---|---|
| APPEND | Add a new row to the database |
| INDEX | Construct an HNSW index using Sim-Encoder embeddings for efficient approximate nearest neighbor search |
| SEARCH | Given an embedding vector $e$, return the corresponding row IDs $k_1..k_n$, seeing Malkov and Yashunin [2018] |
| GET | Given a row ID $k$, retrieve relevant data values, such as actions, Q values, etc. |
| TRIM | Remove historical data to maintain a database size of up to $M$ rows |
| UPDATE | Given a column of data, update the corresponding column in database |

## 3.2 Memory Module

The explainable memory module is in the form of a key-value database. And the keys in the database correspond to the observation embedding vectors obtained via the Sim-Encoder, and each key is associated with a corresponding value that includes information such as the current step's observation, action, reward, and all of other relevant data. To manage this database, we have defined 6 standard operations, namely APPEND, TRIM, GET, UPDATE, INDEX, and SEARCH, which are detailed in Algorithm 1. and Table.1.

The GET operation requires a target state's embedding as the key and returns the corresponding values. To prevent the database from becoming excessively large, we define a TRIM operation that automatically removes older data, retaining only the most recent M records. This design enables efficient storage and retrieval of data while ensuring that the database remains manageable and up-to-date.

In our approach for effective memory retrieval, Approximate K-Nearest Neighbors Search (AKNN) plays a crucial role. We introduce a SEARCH operation that takes a state embedding as input and returns the corresponding key(s) of the nearest neighbor(s) in the database. Additionally, we define an INDEX operation, activated when the database undergoes modifications, seeing Malkov and Yashunin [2018]. This operation reorganizes the HNSW index to align with the updated database, ensuring that subsequent KNN searches remain both fast and accurate.

## 3.3 Policy Evaluation

We introduce the Approximate Nearest Neighbor Search Q-Learning method. In contrast to conventional tabular Q-Learning, we employ a novel form of state value estimation, $\hat{V}(s_t)$, by aggregating the Q-values from the nearest neighbors of the state (cf. Fig.2). The Q-value of each state-action pair is updated following the Bellman equation, incorporating a decay factor, $\gamma$.

During the practical training process, we adopt a batch updating strategy wherein we simultaneously compute the labels for all neighbors of each state and estimate the values of all states in memory. Subsequently, we update all Q-values in the table accordingly. The learning iteration persists until the maximum change in Q-values falls below a specified threshold.

$$q^1..q^N = GET_q(SEARCH(G_\theta(s_t))) \tag{5}$$

$$\hat{v}(s_t) = \frac{1}{N} \sum_{n=1}^{n=N} q^n \tag{6}$$

$$\hat{q}(s_t, a_t) = r_t + \gamma \hat{v}(s_{t+1}) \tag{7}$$

## 3.4 Policy Improvement

For policy improvement, our proposed method directly selects the action with the maximum Q-value from the neighbors ($e^n \in \mathbb{E}, a^n \in \mathbb{A}, q^n \in \mathbb{Q}$), as shown in Equation 10. Using the embedding $e_t$

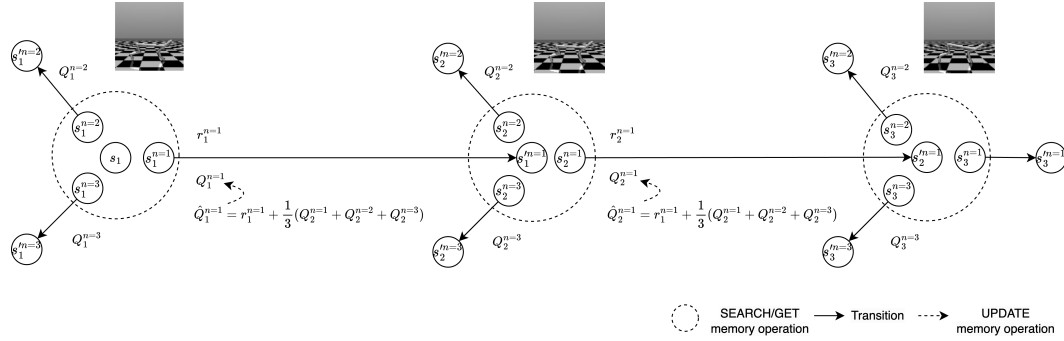

Figure 2: Policy evaluation in memory (N=3)

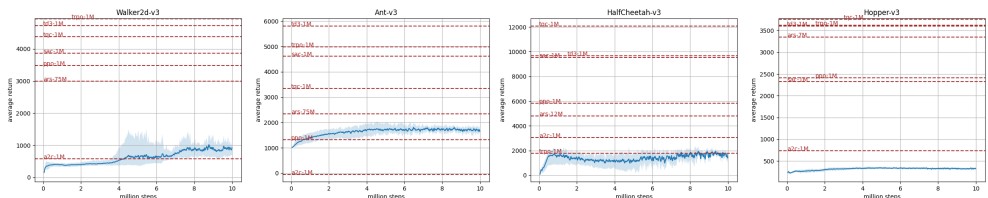

Figure 3: Performance on Continuous Control Tasks vs Conventional RL

generated by the contrastively learned Sim-Encoder, candidates are retrieved from the memory module via the nearest neighbor search. To encourage exploration, action noise following a Gaussian distribution $\mathcal{N}(\mu, \sigma)$ is added.

$$a^1, q^1..a^n, q^n = GET_{aq}(SEARCH(e_t)) \tag{8}$$

$$n_q = argmax_n(q^1..q^n) \tag{9}$$

$$a_t = a^{n_q} + \mathcal{N}(\mu, \sigma) \tag{10}$$

In contrast to employing a black-box network as an actor, we have devised a data-driven, self-explaining actor that seamlessly integrates the results of model search and generates decisions directly using a rule-based approach.

## 4    Experiments

In these experiments, we aimed to evaluate the performance of our ANQ approach in solving continuous control tasks in Mujoco, provide action explainability, and investigate the significance of the Sim-Encoder module in the ANQ framework.

### 4.1    Solving Continuous Control Task in Mujoco

First of all, our approach is evaluated on several continuous control tasks in the MuJoCo physics engine. Specifically, we compare our method with state-of-the-art reinforcement learning (RL) algorithms, including SAC-1M, PPO-1M, and TRPO-1M, on the Walker2d-v3, Ant-v3, HalfCheetah-v3, and Hopper-v3 environments. We use the benchmark performance reported by stable-baselines3 Raffin et al. [2021].

The results (cf. Fig.3) show that our method slightly outperforms A2C-1M on the Walker2d-v3 task and PPO-1M on the Ant-v3 task while achieving comparable performance to TRPO-1M on the

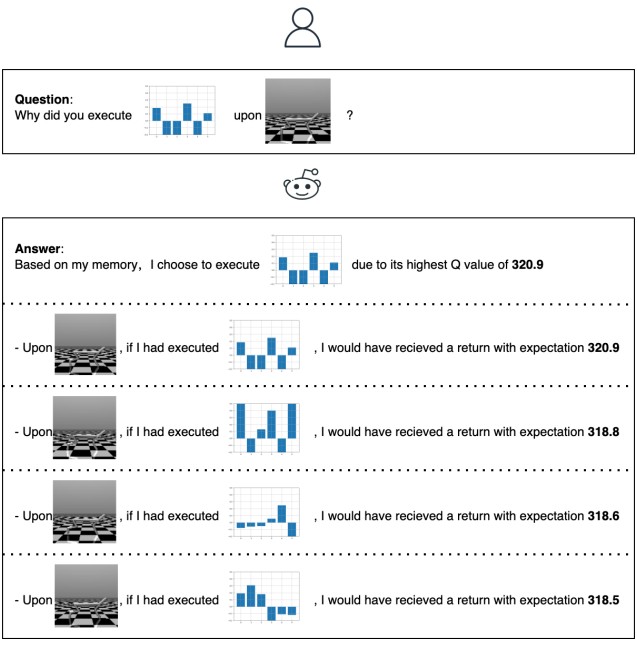

Figure 4: Explainable Action

HalfCheetah-v3 task. Furthermore, we analyze the performance of our method on the Hopper-v3 task by examining the game replay. We find that the agent fails to take the second step and falls after the first step. This indicates that our method may currently lack exploration capability. This will be addressed in future research in order to surpass the performance of traditional RL methods. Overall, our approach successfully and stably converges on the MuJoCo continuous control tasks, but further improvement is necessary to achieve better performance, seeing the discussion in Sec.6.

For the hyperparameters, we utilized a 4-layer MLP network with layer normalization as the encoder. The learning rate was set to 0.0003, the batch size was 512, and the Adam optimizer was used. The size of the explainable memory was limited to 500,000, and old data were discarded once this limit was exceeded. We set the parameters of HNSW to M=16 and ef=10. The total number of training steps was 10 million, and the agent performed ANQ learning every 40,000 environment interactions. We set the number of neighboring actions sampled during each action selection to 10.

## 4.2  Action explainability

In this explainability experiment, we designed a question-and-answer (QA) case (cf. Fig.4) to simulate a scenario where humans need to double-check the correctness of the robot's decision during human-robot collaboration. Specifically, humans ask "why" questions to query the basis of the robot's action, and the robot responds with the policy that it has chosen, as well as the evidence supporting its decision.

To provide a convincing explanation, the robot searches its memory for similar states and explains to the human the actions that it had taken in the past in similar scenarios, as well as the corresponding returns. By providing such detailed explanations, the robot is able to offer valuable insights to humans and effectively bridge the gap in understanding between human and machine decision-making processes, for ensuring safe and reliable human-robot collaboration.

## 4.3  Sim-Encoder

We conducted experiments to investigate the significance of the Sim-Encoder module within the ANQ framework. We have illustrated the retrieved samples (cf. Fig.5). Without the Sim-Encoder, semantically similar states do not share relevant information in cosine space, as discussed in Su et al. [2021]. Our ablation study (cf. Fig.6) demonstrated that the Sim-Encoder led to substantial performance improvements across all four tested tasks, as it effectively retrieves and embeds temporally

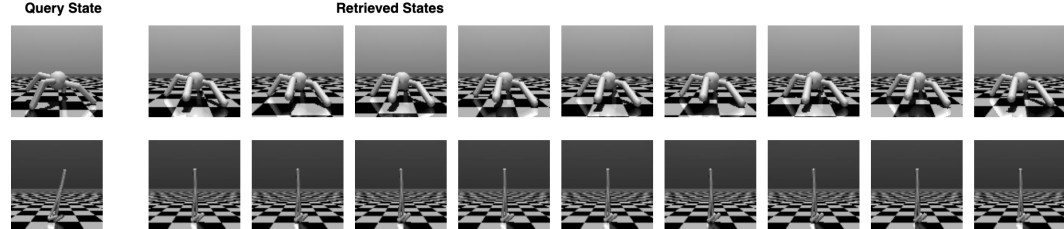

Figure 5: Retrieved Results using Sim-Encoder and Approximate Nearest Neighbor Search

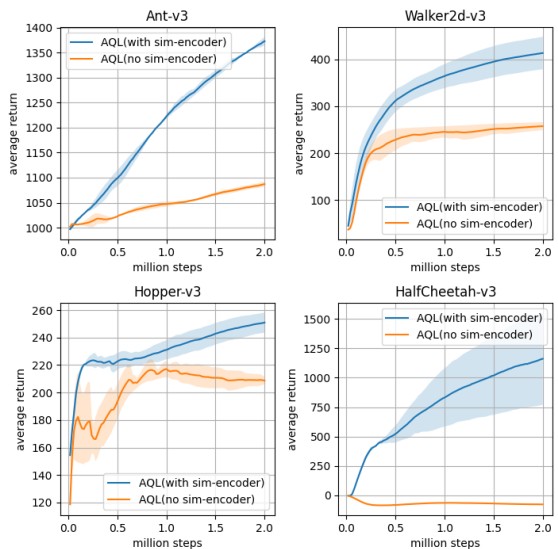

Figure 6: Ablation Study of Embedding Module Sim-encoder

proximate states into a space with adjacent cosine distances. Overall, the Sim-Encoder is an essential component of the ANQ framework and holds potential for use in other RL algorithms.

# 5 Related Work

## 5.1 Episodic Control

The idea of episodic control (EC) was bio-inspired by the mechanism of the hippocampus Lengyel and Dayan [2007]. EC, as a non-parametric approach, possesses virtues including rapid assimilation of past experiences and a solution for sparse-reward situations. Notable works like MFEC Blundell et al. [2016] and NEC Pritzel et al. [2017] employed kNN search to acquire the value for the current state derived from similar states. The value function is in tabular form and updated using the classical Q-learning method. While MFEC adopts random projection Johnson [1984] and VAE Kingma and Welling [2013] as state embedding methods, NEC employs a differentiable CNN encoder instead.

Beyond that, Lin et al. proposed EMDQN Lin et al. [2018], which is a synergy of EC and DQN. Their approach combined the merits of both algorithms, i.e., fast learning at an early stage and good final performance. ERLAM then further promoted the efficacy by introducing an associative memory graph Zhu et al. [2020].

## 5.2 Retrieval-based Learning

The retrieval-based learning and inference architecture provides a viable solution for managing an explainable and extensible knowledge base. One prominent instantiation of this architecture is the retriever-reader model Zhu et al. [2021], which has gained traction in the open domain question answering (openQA) research community. The retriever component returns a set of relevant articles, while the reader extracts the answer from the retrieved documents. Numerous natural language processing (NLP) algorithms, including kNN-LM Khandelwal et al. [2019], RAG Lewis et al. [2020] and RETRO Borgeaud et al. [2022], leverage a retrieval-based approach to enhance their performance and efficiency. These techniques have proven to be effective in the domain of NLP and continue to be an active area of future NLP research Liu et al. [2023].

## 5.3 Explainable Reinforcement Learning

The methods for explainability in reinforcement learning can be broadly categorized into three groups, as discussed in Milani et al. [2022]: Feature Importance (FI), Learning Processing and Markov Decision Process (LPM), and Policy-Level (PL). FI methods involve utilizing decision tree models for explainability, learning an explainable surrogate network through expert and learner frameworks, or directly generating explanations through natural language or saliency maps. LPM addresses explainable transition models to answer "what-if" questions, interpretation of Q values, and identification of key training points. PL provides an understanding of long-term behavior and summarizes the policy. However, many existing explainable reinforcement learning methods require additional network training Guo et al. [2021] or the use of decision trees Silva et al. [2019]. These methods can also impose a cognitive burden on users to understand the model's behavior Dodge et al. [2021]. In contrast, the memory-based reinforcement learning algorithm, ANQ, presented in this paper provides self-explainability without additional explanation specifically training.

# 6 Limitation

In this study, we present an innovative and explainable architecture, termed ANQ, which, despite its novelty, does not significantly outperform state-of-the-art benchmarks. Our primary aim is to demonstrate the efficacy of ANQ with its highly interpretable policy. We acknowledge this performance gap and recognize that our method has not yet incorporated the latest techniques, such as maximum entropy learning from SAC Haarnoja et al. [2018], etc. These refinements will be addressed in future work, rather than here. Moreover, we have not compared our approach with other contrastive learning methods for representation learning. Since we proposed the Sim-Encoder, a thorough comparison with alternative methods and further study will also be included in future research.

# 7 Conclusion

Explainability is crucial in specific domains of reinforcement learning, such as autonomous driving, quantitative trading, and healthcare. To address this challenge, we propose ANQ, a novel semi-parametric reinforcement learning framework that combines the high performance of neural networks with the explainability of a memory-based structure. Additionally, we validate the effectiveness of Sim-Encoder, a key module of ANQ, in state representation and learning efficiency enhancement. Empirical evaluations demonstrate ANQ's effectiveness in solving continuous tasks and providing explainable decision-making. Our contributions include proposing a framework that achieves both efficient control and robust explainability. While further improvements are necessary for superior performance, our results indicate that ANQ is a promising approach for developing explainable and trustworthy RL models in critical applications.

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
