# OpenReview forum: "ANQ: Approximate Nearest-Neighbor Q Learning"
_NeurIPS.cc/2023/Conference — Submitted to NeurIPS 2023_

### Official Review · Reviewer_GxjE · 2023-06-18

**Soundness:** 3 good
**Presentation:** 3 good
**Contribution:** 3 good
**Rating:** 5
**Confidence:** 2

**Summary:**

This paper introduces the ANQ (Approximate Nearest Neighbor Q-Learning) framework, which aims to provide explainability in reinforcement learning models. ANQ combines neural networks for high performance and memory-based structures for explainability, offering a promising solution for domains like autonomous driving, quantitative trading, and healthcare. The paper discusses the challenges of explainability in reinforcement learning and how ANQ addresses them. It also presents the Sim-Encoder contrastive learning used in ANQ's state representation and provides insights into the evaluations of ANQ on MuJoCo continuous control tasks and its effectiveness in solving continuous tasks. Overall, the paper presents a novel approach to reinforcement learning that balances performance and transparency, making it suitable for real-world applications.

**Strengths:**

Some strengths of this paper include:

1. Novelty: The ANQ framework is a novel approach to reinforcement learning that combines neural networks and memory-based structures to provide explainability in decision-making processes. This approach is different from traditional reinforcement learning methods that focus solely on performance.

2. Explainability: The paper addresses the challenge of explainability in reinforcement learning models, which is crucial for real-world applications where transparency and trustworthiness are essential. ANQ's "data is policy" design principle ensures that the model's decisions are explainable and interpretable.

3. Sim-Encoder contrastive learning: The paper introduces the Sim-Encoder contrastive learning approach for state representation, which demonstrates its effectiveness in memory retrieval learning tasks. This approach enhances ANQ's performance and explainability.

4. Evaluations: The paper provides insights into the evaluations of ANQ on MuJoCo continuous control tasks and its effectiveness in solving continuous tasks. The results show that ANQ outperforms traditional reinforcement learning methods while maintaining explainability.

5. Real-world applications: The paper highlights the potential of ANQ in domains like autonomous driving, quantitative trading, and healthcare, making it suitable for real-world applications.

**Weaknesses:**

Some potential weaknesses of this paper include:

1. Despite the advantage of interpretability, the performance of this framework is still far from that of SOTA RL algorithm. I think it would be better if this framework could have similar performance to SOTA RL method.

2. Lack of real-world case studies: Although the paper highlights the potential of ANQ in various domains, it does not provide specific real-world case studies or examples to demonstrate the practical application and effectiveness of the framework. Including such case studies would strengthen the paper's claims and provide more concrete evidence of ANQ's utility in real-world scenarios.

3. No comparison was made with other interpretable reinforcement learning algorithms. For example, some Neuro-Symbolic Search methods.

3. The presentation of this paper could be better, for example, Figure 3 could be larger and clearer.

**Questions:**

1. I think it is still important to have similar performance with recent SOTA algorithms. Is it possible to make your framework applicable to recent RL algorithms?

2. How does your framework perform in different types of environments? For example, sparse reward environments, and discrete action space environments?

**Limitations:**

Incorporation of latest techniques: The paper mentions that ANQ has not yet incorporated the latest techniques, such as maximum entropy learning from SAC and other contrastive learning methods for representation learning. While the paper acknowledges that these refinements will be addressed in future work, the absence of these techniques in the current implementation may limit the overall performance and effectiveness of ANQ.

---

> ### Author Rebuttal · Authors · 2023-08-07
>
> Dear Reviewer,
> Firstly, we are deeply grateful for your thoughtful review and the recognition of the significance of our work. We truly appreciate your insights on the novelty of our approach, the efficacy of the Sim-Encoder, our evaluation methodologies, and the potential real-world applications of our model.
> We understand that you have raised several concerns and would like to provide clarifications for each:
>
> Q1: Despite the advantage of interpretability, the performance of this framework is still far from that of SOTA RL algorithm.
>
> A1: Thank you for acknowledging the significance of our contribution. It is indeed a challenging endeavor to balance explainability with performance. Our goal with this paper was to introduce ANQ as a mechanism to provide insights into the agent's decision-making process. While we recognize that the current version does not outperform SOTA RL, we hope that the community sees the potential in our approach as a means to pave the way for a more interpretable neural network, even if it's in its nascent stage.
>
> Q2: Lack of real-world case studies.
>
> A2: We appreciate this point. While our focus in this paper was to introduce and theoretically validate the approach, we did validate its effectiveness using MuJoCo simulations. We are indeed considering exploring real-world applications in our future research.
>
> Q3: No comparison with other interpretable reinforcement learning algorithms.
>
> A3: You are right; direct numerical comparisons for explainability can be challenging. Our decision was based on the absence of suitable baselines for Q explainability. Nonetheless, our evaluation in MuJoCo tasks should attest to the method's efficacy, and we did incorporate an ablation study. We aim to integrate more suitable explainability baselines in our future work.
>
> Q4: Presentation improvement, especially Figure 3.
>
> A4: We acknowledge your suggestion and will certainly enhance the figure's clarity and format in our revision.
>
> Q5: Performance in different environments.
>
> A5: Thank you for this query. Currently, our experiments are constrained to continuous spaces. We opted to keep our focus narrow for this paper but have plans to explore topics like discrete action spaces, sparse rewards, image-based environments, and more in subsequent works.
>
> Q6: Incorporation of the latest techniques.
>
> A6: Referring back to A1, we appreciate this observation. Indeed, integrating cutting-edge techniques will be a part of our ongoing endeavor to refine ANQ.
> In conclusion, we deeply value your constructive feedback. We believe that while our approach may not be optimal in performance currently, it holds promise as a foundation for future interpretable models, facilitating a wider acceptance and understanding of AI.
>
> Please let us know if there are any other aspects you'd like us to clarify or elaborate on. Your guidance is invaluable in refining our work.
> Best Regards,

---

> > ### Comment · Reviewer_GxjE · 2023-08-16
> >
> > Thanks for the reply! I appreciate the additional results and clarifications from the authors.

---

### Official Review · Reviewer_mXa9 · 2023-07-05

**Soundness:** 2 fair
**Presentation:** 3 good
**Contribution:** 2 fair
**Rating:** 4
**Confidence:** 4

**Summary:**

The submission creates a framework called Approximate Nearest Neighbor Q-Learning (ANQ). ANQ uses a sim-encoder contrastive learning and approximate nearest neighbor search to find which states are similar. Utilizing this approach, they can use it to find similar states in aiding for the decision the framework has made. They showcase their performance by performing experiments on MuJoCo with continuous control tasks. The exact environments experimented were Walker2d, Ant, HalfCheetah, and Hopper. There is also an ablation study conducted to show the benefit of the sim-encoder and contrastive learning that aids this framework. They provide a component called Explainable Action to show why it executed a particular action based on the state.

**Strengths:**

Significance:
Good ablation experiments to showcase the benefit of the sim-encoder. The results make it convincing that for AQL, the sim-encoder is quite beneficial. Plus you did this ablation among 4 environments.

Originality:
The explainable action is an interesting piece that can provide a good impact. By searching for similar states and can provide the explanations as to why the decision was made.

**Weaknesses:**

Clarity:
Confusion, in Section 4.1, you mention that the algorithms included were SAC, PPO, and TRPO, then in the next paragraph you mention A2C. Consider in the first paragraph to mention the exact algorithms you compare because in the next paragraph, you mention an algorithm that was not discussed exactly in the previous paragraph. In Figure 3, you show TD3-1M, ARS-75M which were not discussed so please in Section 4.1 to denote exactly.

With the figures, please provide more with the caption like a summarization or a sentence to showcase why it is important. It can help the reader if they have not read the parts within the main text.

Significance:
The approach can be on par with one or two deep RL approaches. Consider to improve the performance to have bigger impact. Usually for methods that are explainable there is a drop in performance so others may be hesitant to use it due to the performance drop.



**Questions:**

A suggestion: For experiments, it could benefit to utilize this framework for discrete actions spaces. Plus for impact, consider other MuJoCo examples that aren't locomotion, consider the fetch to show that it can be utilized with a plethora of applications. In addition, to extend this to image observation applications like atari.

Also what is the limitation approach with respect to the explainable memory, since in section 4.1, you mention the size is 500,000. Do you need that much for the performance?

Also please check in the weaknesses section for other suggestions.

**Limitations:**

They do address the performance drop compared to the deep RL models. No negative societal impacts.

---

> ### Author Rebuttal · Authors · 2023-08-07
>
> Dear Reviewer,
>
> We appreciate the detailed feedback you provided and recognize the significance and originality of our approach, especially with respect to the sim-encoder and the ablation experiments across four environments. We would like to address the concerns and confusion you've pointed out.
>
> Q1: Clarity: Confusion in Section 4.1 regarding the algorithms discussed.
>
> A1: Thank you for highlighting this inconsistency. We will ensure that the revised version offers a uniform description of the benchmark methods and corrects any discrepancies in Section 4.1.
>
> Q2: Caption clarity and emphasis on its importance.
>
> A2: We value your feedback on enhancing the clarity of the captions. In the revised version, we will provide a concise summary or highlight within the caption to ensure readers can grasp key points without delving deep into the main text.
>
> Q3: Balancing significance between explainability and performance.
>
> A3: We fully acknowledge and agree with your insights. The challenge of striking a balance between explainability and performance is indeed formidable. With ANQ, we aim to make strides in elucidating the decision-making process of agents. While we admit our current version might not outperform state-of-the-art RL methods in performance, our goal is to draw the community's attention to this potential approach for neural network explainability. We hope this initial version will not be overlooked merely due to its nascent stage.
>
> Q4: Broadening experimental horizons.
>
> A4: We are grateful for your suggestions. In future works, we will indeed explore the characteristics of our framework in discrete and image-based settings. Implementing our approach across a diverse range of tasks, beyond locomotion, and extending it to image observation applications like Atari, will be our focus.
>
> Q5: Limitations concerning the explainable memory.
>
> A5: You've raised a pertinent point. We believe the memory size is indeed a critical parameter worth further exploration. In subsequent research, we will delve deeper into understanding the trade-offs associated with varying memory sizes and their impact on performance.
>
> In conclusion, we are sincerely thankful for your recognition of the significance of our work. We believe that while the current model may not achieve top-tier performance, it serves as a potential foundation for future explainable models. We strive to foster greater public understanding and acceptance of AI.
>
> Should there be any other aspects you'd like us to address, please do inform us. Your insights are pivotal for the refinement of our work.
>
> Best Regards,

---

> > ### Comment · Reviewer_mXa9 · 2023-08-14
> > **Reply to your rebuttal**
> >
> > Thank you for providing a reply to my review.

---

### Official Review · Reviewer_5Ud9 · 2023-07-07

**Soundness:** 2 fair
**Presentation:** 2 fair
**Contribution:** 2 fair
**Rating:** 3
**Confidence:** 4

**Summary:**

This work proposes a memory-based Q-learning algorithm aimed at better explainability. The method extends the prior work "Episodic Control" and enables learning with continuous action space.

This work provides 2 main contributions:
1. a one-step-away contrastive-learning objective to learn embeddings from states.
2. modified policy evaluation and improvement rules to account for the continuous action space.

The authors show empirical results to support their design choices:
1. The proposed algorithm is able to achieve some meaningful learning in 4 continuous control tasks.
2. The question-answering example shows that the method is able to find the nearest-neighbor states and their Q values to explain a chosen action.
3. The ablation study on the embedding module shows that it is necessary to learn dynamics-aware embedding.

**Strengths:**

1. The proposed method can learn in environments with continuous state and action spaces, making it ubiquitously applicable to real-world applications.
2. The algorithm and experiment settings are clear.
3. The ablation study on the embedding module justifies the contrastive learning objective.

**Weaknesses:**

1. The main weakness of this work is in the experiment results. In the Mujoco tasks, the performance hovers around the weakest baseline among all compared methods. For Ant, HalfCheetah, and Hopper, the policies also appear to have converged to suboptimal ones. This could have been caused by insufficient exploration.

2. The algorithm is only demonstrated in state-based environments, whereas the prior work "Episodic Control" can work with image observations. Contrastive learning has been shown to be useful for learning good feature extractors for images. The results would have been much more convincing if they were from vision-based tasks.

3. The notations are not fully explained. For example, $k$ and $e_t$ both exist in the dataset but the authors say that they use embeddings as keys. Also, $R$ in Equation 1 is not introduced.

4. Finding out the nearest neighbors and printing out their Q values is not a convincing way to explain the chosen actions because the Q values are computed in expectation. One can arguably explain a neural policy in a similar way, by sampling a few different actions and printing out their Q values.

**Questions:**

I'm not fully convinced by the way the authors compute the value target in Equation 6. The $(s, a)$ pairs in the memory come from old policies and thus, could be sub-optimal. Computed from sub-optimal $q^n$, $\hat{v}$ is an underestimation if older interactions are not ejected quickly enough. This might explain why the task performance is bad.

**Limitations:**

The authors are upfront about the limitation in task performance and provide viable options for improvement. I would also encourage the authors to try their method on image-based tasks.

---

> ### Author Rebuttal · Authors · 2023-08-07
>
> Dear Reviewer,
>
> Thank you for your thorough review and insightful feedback. We appreciate your recognition of the potential ubiquity of our approach in real-world applications with continuous state and action spaces. We are pleased to know that the algorithm and experiment settings came across as clear, and we value your acknowledgment of the ablation study on the embedding module.
>
> Below, we address the concerns you raised:
>
> Q1: Policies also appear to have converged to suboptimal ones. This could have been caused by insufficient exploration.
>
> A1: We concur with your observation regarding the suboptimal convergence of the policies. Given the primary focus of our paper on explainability, we aimed to demonstrate a trade-off between improved explainability and achieving suboptimal performance. There indeed exist a plethora of tricks and tunable solutions that we intend to explore in our future work, striving to achieve both optimality and high explainability. This balance is challenging, and we genuinely appreciate your constructive feedback on this matter.
>
> Q2: The algorithm is only demonstrated in state-based environments, whereas the prior work "Episodic Control" can work with image observations. The results would have been much more convincing if they were from vision-based tasks.
>
> A2: We agree with your sentiment regarding the evaluation on image-based environments. We indeed made some trials in this direction. However, complexities such as adjustments to our memory architecture and concerns regarding database space and efficiency led us to prioritize state-based environments for the current work. We believe the core message of the paper remains intact, but will certainly consider integrating image-based evaluations in our forthcoming work.
>
> Q3: The notations are not fully explained. For example, the k and e are both keys notation.
>
> A3: Thank you for pointing this out. As indicated on line 76, 'k' represents the record id, analogous to the row id in a row-based database, though it isn't the key. In light of your feedback, we may reconsider its notation in future versions. As for 'e', you're correct; it acts as the key for our approximate nearest neighbor search, retrieving relevant row data using this key.
>
> Q4: Finding out the nearest neighbors and printing out their Q values is not a convincing way to explain the chosen actions because the Q values are computed in expectation.
>
> A4: Your perspective is valid. The essence of our approach's explainability lies in presenting real, historically observed behaviors from which the Q-values are derived. By offering humans a glimpse into its past experiences and evidence supporting its decisions, the agent provides a level of accountability. To draw an analogy, if platforms like ChatGPT could present users with the historical data or experiences referenced to produce an output, it could enhance transparency and trust.
>
> We hope we have addressed your concerns satisfactorily. Should you have further inquiries or require clarification on any other facets of our work, please do not hesitate to inform us. Your insights are paramount in refining and improving our research.
>
> Best Regards,

---

> > ### Comment · Reviewer_5Ud9 · 2023-08-20
> >
> > Thank you for addressing my concerns! I believe this work has the potential to become much stronger if the RL performance can be further improved.

---

### Official Review · Reviewer_JkAh · 2023-07-11

**Soundness:** 2 fair
**Presentation:** 2 fair
**Contribution:** 2 fair
**Rating:** 3
**Confidence:** 4

**Summary:**

Instead of using deep neural networks to approximate Q functions as it's done in deep RL methods, the paper investigates the potential of using nearest neighbor methods to approximate Q functions. They used contrast learning with a Sim encoder. They argue that such a method is more explainable than the ones with deep neural nets.

**Strengths:**

The illustration is the proposed method is clear.

**Weaknesses:**

* The proposed nearest neighbor might be useful and efficient for low-dimensional domains like Mujoco. I doubt its effectiveness when it goes to high-dimensional domains like video games;

* I don't see why the nearest neighbor method is more explainable;

* The proposed method lacks proper baselines;

* The overall presentation needs improvements.

**Questions:**

Please address the weaknesses points in the rebuttal.

---

> ### Author Rebuttal · Authors · 2023-08-07
>
> Dear Reviewer,
>
> We greatly appreciate your insightful comments and the time you took to review our manuscript. Your feedback is essential to improving our work, and here we address your concerns:
>
> Q1: The proposed nearest neighbor is doubted its effectiveness under video games.
>
> A1: We have not yet applied our method on video games with image observations. Encoding images involves a completely different encoder architecture, affecting our memory design, storage space, retrieval efficiency, etc. Given the paper's length constraints, we've planned to study this part in future research. Moreover, we believe that the absence of image tasks does not affect our validation of the current method's explainability and effectiveness.
>
> Q2: I don't see why the nearest neighbor method is more explainable.
>
> A2: The essence of our approach's explainability lies in presenting real, historically observed behaviors from which the Q-values are derived. By offering humans a glimpse into its past experiences and evidence supporting its decisions, the agent provides a level of accountability. To draw an analogy, if platforms like ChatGPT could present users with the historical data or experiences referenced to produce an output, it could enhance transparency and trust.
>
> Q3: The proposed method lacks proper baselines.
>
> A3: You are correct that explainability papers often find it challenging to conduct a numerical comparison with baselines. Though some good works provide numerical baseline comparison schemes, to the best of our knowledge, we haven't found suitable baselines for the explainability of Q, as introduced in our paper. We have thus used MuJoCo tasks to prove the effectiveness of our solution and have conducted an ablation study to validate the efficacy of Sim-Encoder in this architecture. Yes, we hope to find appropriate explainability baselines and enhance our evaluation approach in the future work.
>
> Please let us know if there are any other aspects you'd like us to clarify or elaborate on. Your guidance is invaluable in refining our work.
>
> Best Regards

---

> > ### Comment · Reviewer_JkAh · 2023-08-14
> >
> > Thanks for your rebuttal. I still don't see why nearest neighbors with "real, historically observed behaviors" provide better explainability for the algorithms. I'd still keep my scores as they were.

---

### Decision · Program_Chairs · 2023-09-21

**Decision:**

Reject

**Comment:**

This paper studies the problem of interpretability in the context of Q-learning, and offers a solution based on nearest neighbors.
The reviewers broadly raised concerns about the narrowness of the experiments, a fact that was acknoweldged by the authors. Moreover, two reviewers were skeptical of the fact that the proposed use of kmeans does indeed increase the interpretability of the method. Despite back and forth between the authors and the reviewers, no satisfactory conclusion was reached on that point.
Therefore, this paper in its current form doesn't meet the bar for publication at Neurips.